# Rekindling Vision: Innovative Strategies for Treating Retinal Degeneration

**DOI:** 10.3390/ijms26094078

**Published:** 2025-04-25

**Authors:** Irfan Khan, Faiza Ramzan, Haroon Tayyab, Karim F. Damji

**Affiliations:** 1Department of Ophthalmology and Visual Sciences, The Aga Khan University, Stadium Road, P.O. Box 3500, Karachi 74800, Sindh, Pakistan; haroon.tayyab@aku.edu; 2Centre for Regenerative Medicine and Stem Cells Research, The Aga Khan University, Stadium Road, P.O. Box 3500, Karachi 74800, Sindh, Pakistan; 3Department of Biological and Biomedical Sciences, The Aga Khan University, Stadium Road, P.O. Box 3500, Karachi 74800, Sindh, Pakistan; 4Dr. Panjwani Center for Molecular Medicine and Drug Research, International Center for Chemical and Biological Sciences, University of Karachi, Karachi 75270, Sindh, Pakistan; faizakhanmeo@gmail.com; 5Department of Ophthalmology and Visual Sciences, University of Alberta, Edmonton, AB T6G 2R3, Canada

**Keywords:** retina, vision, stem cells, gene therapy, extracellular vesicles, molecules, vision restoration

## Abstract

Retinal degeneration, characterized by the progressive loss of photoreceptors, retinal pigment epithelium cells, and/or ganglion cells, is a leading cause of vision impairment. These diseases are generally classified as inherited (e.g., retinitis pigmentosa, Stargardt disease) or acquired (e.g., age-related macular degeneration, diabetic retinopathy, glaucoma) ocular disorders that can lead to blindness. Available treatment options focus on managing symptoms or slowing disease progression and do not address the underlying causes of these diseases. However, recent advancements in regenerative medicine offer alternative solutions for repairing or protecting degenerated retinal tissue. Stem and progenitor cell therapies have shown great potential to differentiate into various retinal cell types and can be combined with gene editing, extracellular vesicles and exosomes, and bioactive molecules to modulate degenerative cellular pathways. Additionally, gene therapy and neuroprotective molecules play a crucial role in enhancing the efficacy of regenerative approaches. These innovative strategies hold the potential to halt the progression of retinal degenerative disorders, repair or replace damaged cells, and improve visual function, ultimately leading to a better quality of life for those affected.

## 1. Introduction

The layer deep inside the eye that detects light is called the retina. Photoreceptor cells within the retina are involved in the perception of light. Underlying the photoreceptors, the retinal pigment epithelial cells (RPE) actively participate in phototransduction. There is also a complex interaction of various cell types within the retina, culminating in electrical impulses that are sent via the retinal ganglion cells (RGCs) through the optic nerve into the brain. Visual loss can occur in different degrees depending on the malfunctioning of these three major cell types [1,2]. The degeneration of RGCs is the hallmark of glaucoma, and the degeneration of photoreceptors and/or RPE is responsible for retinal degenerative conditions. These can be inherited or acquired (e.g., age-related macular degeneration, diabetic retinopathy). The hereditary type, often referred to as retinitis pigmentosa (RP), affects around 1 in 3000 persons globally [3]. Neural cell death is the result of retinal degeneration, a complex process involving several factors and signaling pathways. While the hereditary and complicated types of retinal degeneration exhibit unique characteristics, they share several signaling cascades, including inflammation and oxidative stress. During the past decade, the cause underlying the disease and novel treatments have been intensively investigated.

The objective of this review is to highlight pathophysiological insights and evaluate novel treatment approaches for various retinal degenerative conditions (RDCs). Conventional therapies for RDCs are limited to symptom management and halting or slowing disease progression; they typically do not address the root causes of the condition. To replace lost or damaged retinal tissue for patients with RDCs, regenerative methods, especially those incorporating stem and progenitor cell therapies, offer a paradigm shift in management [4,5,6]. These cutting-edge treatments use stem cells’ ability to develop and mature into multiple types of retinal cells to replenish the cell population(s) required for vision [5,6]. Furthermore, these approaches have been strengthened by progress in gene editing, extracellular vesicles, and exosome technologies, which improve repair processes and enable targeted therapeutic administration. Gene therapy and small molecules further enhance these regeneration techniques, which can ameliorate genetic defects and modify cellular pathways. When combined, these methods have the potential to arrest and reverse damage and restore visual function, leading to enhanced quality of life for those who suffer from these crippling disorders [7,8,9,10].

## 2. Cell Therapy for Retinal Diseases

Future efforts to manage retinal degenerative illnesses may entail the management of inflammation and support for the surviving retinal cells, as most of these illnesses are caused by the loss of specialized retinal cells and local inflammatory events. Several types of stem cells may have paracrine effects that lessen inflammation and prolong the life of the surviving retinal cells [11]. An option for therapy is pluripotent stem cells with remarkable differentiation potential. An alternative approach is induced pluripotent stem cells (iPSCs) by reprogramming normal somatic cells, which can be matured into retinal cell types [12,13]. However, the use of iPSCs and embryonic stem cells (ESCs) is restricted due to the possibility of immunological rejection, teratogenicity, and ethical considerations in the case of ESCs. These factors suggest that mesenchymal stem cells (MSCs) hold promise and can be used to treat retinal degeneration. To use MSCs as autologous cells without immunological rejection, the patient’s bone marrow or adipose tissue can be used to harvest and cultivate these cells in vitro. The long-term viability of MSCs following injection into the vitreous body has been demonstrated; they may also protect RGCs and encourage the regeneration of axons after compression of the optic nerve [14,15,16]. MSCs are relatively easy to extract, isolate, and grow in sufficient quantities from a range of tissues in vitro for autologous or allogenic applications. It has been shown that MSCs retain their ability to differentiate into a number of cell types, including those that express RPE or photoreceptor cell markers [17]. The well-documented characteristics of MSCs include their anti-inflammatory potential and their capability to facilitate ocular surface repair [18,19]. MSC transplantation does not seem to pose a risk for tumor formation in animal models [18].

### 2.1. Characteristics of MSCs

MSCs are adult stem cells that were first identified as a population of bone marrow-derived cells that adhere to plastic surfaces and organize into colonies that resemble fibroblasts. These cells have several advantages, including the ability to be used as autologous cells, excellent growth potential during in vitro propagation, and relatively straightforward and easy isolation from bone marrow or adipose tissue, umbilical cord, and other adult tissue sources [19]. MSCs have the ability to differentiate into distinct cell types of the mesenchymal line [20,21,22,23,24,25,26]. MSCs can develop into osteoblasts, chondroblasts, and adipocytes and express the surface markers CD105, CD73, and CD90 positively, but they lack the expression of CD45, CD34, CD14, CD19, and CD11b [27,28,29,30,31]. It has been shown that MSCs produce a wide range of growth factors and cytokines, possess potent anti-inflammatory and immunomodulatory properties, and support tissue regeneration.

#### 2.1.1. Other Types of Stem Cells

The capacity of iPSCs to develop into a diverse range of retinal cell types, such as photoreceptors, RPE, and RGCs, has attracted much attention. Autologous transplantation may be possible with iPSCs since adult somatic cells can be reprogrammed to a pluripotent state, encountering the barrier of immunological rejection [20,21]. Furthermore, the generation of RPE cells and the treatment of retinal degenerative illnesses have demonstrated the beneficial use of ESCs [20,21]. Notwithstanding the difficulties these options present, such as possible tumorigenicity and ethical issues with ESCs, they must be included in the conversation to provide a thorough understanding of the changing field of retinal regeneration treatments through the integration of neural stem cells (NSCs), ESCs, and iPSCs, among other kinds of stem cells [20,21].

#### 2.1.2. Immunoregulatory Properties of MSCs

MSCs are known for their potent immunosuppressive capabilities. They secrete a wide range of immunomodulatory molecules and interact directly with immune cells through cell-to-cell contact, suppressing the function of antigen-presenting cells and activating regulatory T cells. MSCs inhibit the proliferation of lymphocytes in response to mitogens and alloantigens [32], and they have been shown to suppress T and B cell proliferation, cytokine production, and the activity of cytotoxic T and natural killer (NK) cells [33]. In vivo studies reveal that MSC treatments reduce the incidence of graft-versus-host disease (GVHD), prevent corneal allograft rejection, mitigate septic complications, and modulate the occurrence and severity of autoimmune disorders [34]. MSCs exert their immunosuppressive effects through a variety of molecules, including cyclooxygenase-2 (Cox2), indoleamine 2,3-dioxygenase (IDO), TNF-α-stimulated gene 6 protein (TSG-6), programmed death-ligand 1 (PD-L1), and Fas-L molecule [35]. Additionally, MSCs produce several cytokines, such as transforming growth factor-β (TGF-β) and interleukin-6 (IL-6), which play key roles in regulating the balance between proinflammatory Th17 cells and anti-inflammatory regulatory T cells (Tregs) [36]. The ability of MSCs to migrate to sites of inflammation or injury and support tissue regeneration underscores the therapeutic potential of systemic MSC administration [37,38,39].

#### 2.1.3. Anti-Apoptotic Effect of MSCs

Degenerative and inflammatory signaling in retinas is often associated with the localized overproduction of various cytokines. These proinflammatory mediators can be released by activated retinal cells or infiltrating immune cells, contributing to cellular damage. It is shown that elevated levels of proinflammatory cytokines can induce apoptosis in neighboring cells, both in vitro and in vivo [40]. In addition, prolonged inflammation can lead to endoplasmic reticulum stress, which further promotes apoptosis. Proinflammatory cytokines also alter the expression of key apoptotic genes, such as p53, Bax, and Bcl-2, exacerbating cell death.

Chemokines and cytokines produced by immune cells in response to retinal injury attract more immune cells to the site, intensifying inflammation and apoptotic pathways [41]. Modulating local inflammation and inhibiting apoptosis are promising strategies to prevent or minimize retinal damage. MSCs, with their immunoregulatory, anti-inflammatory, and anti-apoptotic properties, offer a potential therapeutic approach for treating retinal diseases [42]. MSCs can reduce apoptosis and suppress the expression of pro-apoptotic genes in corneal explants cultured in the presence of proinflammatory cytokines that typically trigger cell death [43].

#### 2.1.4. Growth Factors Secreted by MSCs

MSCs are potent producers of various trophic and growth factors. While some of these factors are constitutively secreted by MSCs, others are released in response to activation by mitogens, proinflammatory cytokines, or other signals. The primary therapeutic mechanisms of MSCs are thought to involve their ability to secrete growth factors and function via paracrine signaling [32]. These growth factors are believed to play a key role in retinal regeneration and include platelet-derived growth factor (PDGF), hepatocyte growth factor (HGF), fibroblast growth factor (FGF), nerve growth factor (NGF), insulin-like growth factor-1 (IGF-1), glial cell-derived neurotrophic factor (GDNF), epidermal growth factor (EGF), angiopoietin-1, erythropoietin, vascular endothelial growth factor (VEGF), and transforming growth factor-beta (TGF-β) [44].

MSCs can naturally produce some of these factors, but their production can be significantly upregulated in response to proinflammatory cytokine stimulation. However, the presence of certain proinflammatory cytokines can also suppress the synthesis of other growth factors, such as TGF-β and HGF [27]. MSCs differentiate into cells expressing retinal markers, increasing their secretion of specific growth factors. For example, differentiated MSCs exhibit higher levels of NGF, GDNF, and IL-6, suggesting enhanced potential for retinal tissue regeneration. Additionally, research has demonstrated that MSCs increase their production of neurotrophic factors in response to byproducts of light-induced retinal injury, which in turn helps delay apoptosis in damaged retinal cells [45]. Further studies have shown that the secretion of neurotrophic factors by MSCs promotes photoreceptor survival in models of retinal degeneration following subretinal transplantation. In vitro studies also indicate that photoreceptor viability is enhanced by MSCs. These findings suggest that MSCs differentiated into retinal-like cells exhibit greater secretory activity and regenerative potential compared to undifferentiated MSCs [46,47].

#### 2.1.5. MSC Differentiation

A defining characteristic of stem cells is their ability to transdifferentiate or develop into different cell types. While there is substantial evidence regarding the differentiation of MSCs and other stem cells into corneal and other ocular cell types [48,49,50], less is known about their ability to differentiate into specific retinal cells or neurons.

MSCs derived from rat conjunctiva expressed markers associated with photoreceptors and bipolar cells when cultured in the presence of taurine. Other studies have used taurine, along with activin A and epidermal growth factor (EGF), to promote the differentiation of MSCs into photoreceptor-like cells. Under conditions conducive to differentiation, MSCs expressed photoreceptor-specific genes such as Rho and Rlbp after 8–10 days of culture [51,52,53,54,55]. Additionally, MSCs injected into the subretinal space can integrate into the retina and express photoreceptor-specific markers. Subsequent investigations showed that MSCs transplanted into damaged retinas displayed characteristics of photoreceptors, bipolar cells, and amacrine cells [55,56,57]. Other studies have also reported the successful differentiation of MSCs into retinal pigment epithelial (RPE) cells, which are essential for maintaining photoreceptor function. Supernatant from mouse spleen cells activated with concanavalin A combined with retinal cell extract to mimic the environment of an injured retina. When mouse bone marrow-derived MSCs (BM-MSCs) were cultured in these conditions for seven days, they began expressing retinal cell markers, such as rhodopsin, S antigen, retinaldehyde-binding protein, calbindin 2, recoverin, and retinal pigment epithelium 65 (*RPE65*). Interferon-γ, present in the supernatant of activated spleen cells, was identified as a key factor supporting the differentiation of MSCs into retinal-like cells. The differentiated MSCs also produced numerous neurotrophic factors, which are critical for retinal regeneration. This study, along with others [51,52,53], demonstrated that signals from damaged retinas stimulate MSC differentiation into cells expressing retinal markers, and differentiation is enhanced by cytokines produced by activated immune cells [58,59]. Figure 1 shows protective mechanisms of retinal cells mediated by transplanted cells.

#### 2.1.6. Trophic Factors Contributing to the Therapeutic Potential of MSCs

In addition to secreting extracellular vesicles (EVs), MSCs secrete various molecules that promote growth, immunoregulation, and neurotrophic responses. These EVs contain bioactive molecules that support cell viability. Intravitreal injection of EVs can be as effective as MSCs in restoring vision in experimental models of retinal laser injury [21]. A rat optic nerve crush model treated with exosomes derived from MSCs preserved the function of retinal ganglion cells, highlighting the neuroprotective potential of MSC-derived vesicles [60]. Additionally, MSCs can improve anti-inflammatory conditions and enhance cell survival through mitochondrial transfer. This process may have significant implications for the treatment of retinal diseases, as such conditions are associated with mitochondrial dysfunction [61,62,63,64]. MSCs have been shown to possess the ability to fuse with other cell types, including damaged retinal cells in various models. This potential for cell fusion may exert its therapeutic effects when injected intraocularly, enhancing the regenerative impact [61,62,63,64].

#### 2.1.7. Potential of MSCs for the Treatment of Retinal Diseases

Several experimental studies have demonstrated the therapeutic potential of MSCs in treating retinal disorders [65,66,67]. MSC implantation has been shown to significantly slow retinal degeneration, promote the regeneration of cone cells, axons, and RPE, and enhance the survival of RGCs. Based on these promising results, MSCs have been suggested and investigated as a possible treatment for various retinal diseases [68,69]. The potential to use stem cells for treating currently incurable retinal diseases is highly appealing [70,71,72,73]. However, extensive research is needed to refine stem cell-based therapies before they can be widely applied in clinical settings. To advance these therapies, animal models have been developed to simulate retinal disorders, either through genetic modification or pharmacological induction of retinal cell degeneration. MSCs secrete a wide range of EVs that contain growth factors, immunoregulatory agents, and neurotrophic compounds, which may support cell viability. In an experimental model of retinal laser injury, intravitreal injection of EVs was found to be just as effective as MSCs in restoring vision [74,75,76,77,78]. The ability of EVs to mimic MSCs’ therapeutic effects further supports the potential use of stem cell-derived therapies for retinal regeneration. Stem cell therapy offers hope to patients suffering from vision loss for improving their quality of life [79,80,81].

#### 2.1.8. Retinal Regeneration by MSCs

MSCs play a pivotal role in retinal regeneration by orchestrating a complex network of signaling pathways and molecules essential for optimal tissue repair. Primarily through paracrine signaling, MSCs secrete a plethora of growth factors and cytokines that promote retinal cell survival, proliferation, and differentiation. Key growth factors, such as NGF and brain-derived neurotrophic factor (BDNF), support angiogenesis and maintain the viability of retinal cells, respectively. Fibroblast growth factors (FGFs) and glial cell line-derived neurotrophic factor (GDNF) further contribute to retinal cell survival and function by regulating cell proliferation, differentiation, and injury repair [82,83]. Beyond their trophic functions, MSCs exhibit potent anti-inflammatory and immunomodulatory properties, creating a conducive environment for retinal regeneration. Prostaglandin E2 (PGE2), TGF-β, and interleukin-10 (IL-10) collectively suppress inflammatory responses, facilitating tissue healing. Additionally, MSCs release extracellular vesicles and exosomes containing bioactive mediators, including proteins and microRNAs (miRNAs), which regulate gene expression, enhance cell survival, and promote regeneration [84].

MSCs also possess the remarkable capacity to differentiate into various retinal cell types under specific conditions. Retinoic acid (RA) and signaling pathways like notch and Wnt/β-catenin guide cell fate determination, proliferation, and differentiation, leading to the generation of functional photoreceptors and retinal pigment epithelial cells. Furthermore, MSC-secreted factors such as pigment epithelium-derived factor (PEDF) and ciliary neurotrophic factor (CNTF) provide neuroprotection and anti-angiogenic properties, preventing the degeneration of existing retinal neurons. Matrix metalloproteinases (MMPs), produced by MSCs, facilitate tissue remodeling by breaking down extracellular matrix components, a crucial step in efficient regeneration [84].

Understanding and harnessing these intricate pathways and molecular mechanisms, such as PI3K/Akt, MAPK/ERK, JAK/STAT, and HIF-1α pathways, is essential for developing effective MSC-based therapies to enhance retinal regeneration. By targeting specific aspects of the repair process, researchers aim to improve outcomes for patients with various retinal disorders, offering promising avenues for future medical treatments [85]. Figure 2 demonstrates the mechanisms of retinal regeneration through MSCs.

#### 2.1.9. Limitations of MSC-Based Therapy

While both animal and human studies have demonstrated the safety of MSCs therapy, several critical factors must be carefully considered before translating MSC therapy from preclinical models into clinical practice [18,86]. The heterogeneity of MSCs poses a significant challenge due to variations in cell source, isolation methods, and culture conditions. The choice of MSC dosage, administration route, and timing can also influence treatment outcomes. Prolonged cell culture has been shown to affect MSC morphology, secretory capacity, and migratory characteristics [87,88]. Consequently, establishing standardized protocols for MSC preparation is essential for consistent and effective applications. The long-term survival of MSCs in vivo remains a subject of debate. Although MSCs are generally considered immunologically privileged, they can express MHC Class II and costimulatory molecules under certain conditions, potentially attracting immune cells' attention. While several studies have reported the long-term survival of allogeneic or even xenogeneic MSCs in immunocompetent patients, Eggenhofer et al. challenged this notion, suggesting a shorter lifespan for in vitro cultured MSCs. However, MSCs may persist in immunologically favorable environments, such as the eye [89]. The destiny and immunological roles of MSCs, once they are transferred into the inflammatory milieu of the damaged retina, remain an open question. Immunosuppressive MSCs may become a cell population that promotes the growth of aggressive proinflammatory Th17 cells if they are introduced into an environment containing proinflammatory cytokines. The route of MSC administration is another crucial consideration. Intravenous injections often result in a low fraction of MSCs reaching the eye. Therefore, intraocular or periocular administration, such as intravitreal, suprachoroidal, subtenon, or subretinal injection, appears to be more promising for retinal treatment [88,90,91,92]. Addressing these challenges and limitations will advance MSC-based therapies for retinal diseases and ensure their safe and effective clinical application.

Since the eye is an ideal organ for cell-based transplantation, stem cells capable of self-replicating and differentiating into retinal neurons offer a promising future approach. First, the retinal architecture allows the introduction of viable therapeutic cells, such as photoreceptors (PRs) or RPE, due to their potential for expansion [93,94]. Second, the sub-retinal space’s relative immunological privilege helps shield transplanted cells from immune rejection. Third, well-established surgical techniques like vitrectomy with subretinal and intravitreal injections have a strong safety record. Fourth, noninvasive technologies facilitate the monitoring of changes in retinal/optic nerve structure and function [95,96]. These advantages make stem cell-based therapies a practical and affordable option for treating retinal degenerative disorders. Phase I/II clinical studies have demonstrated the feasibility of stem cell-based treatments using bone marrow MSCs, iPSCs, and ESCs [97]. Cells derived from ocular tissues, particularly retinal progenitor cells (RPCs) and retinal stem cells (RSCs), are promising candidates for cell replacement therapies due to their ability to evade immune responses and avoid cancerous transformation. However, RPCs or RSC-based treatments face challenges such as limited growth and survival potential. Addressing these issues requires the development of appropriate cell delivery systems and structured therapeutic delivery substrates for the sub-retinal space.

Scaffolds combined with seeded cells represent a novel approach to treating retinal degenerative diseases. Further investigations are required to optimize these scaffolds and explore the potential benefits of nanotechnology. Nanoscale scaffolds can enhance cell function and improve the precision and efficiency of therapeutic delivery, opening new avenues for advancing retinal regenerative therapies [98]. Table 1 describes the different cell sources and how they are used in the regeneration of the retina.

## 3. Therapeutic Role of Extracellular Vesicles (EVs) and Exosomes in Eye Diseases

Since the discovery of their biological characteristics, EVs have emerged as promising therapeutic agents. Early evidence in ophthalmology involved using a peptide derived from αB-crystallin incorporated into a polymer nanoparticle to protect the RPE under stress [117]. Similarly, tailored liposomes were developed to improve the delivery of lipophilic substances to the retina, addressing challenges associated with ocular medication administration [118,119]. In vitro studies have revealed that exosomes can influence the pharmacokinetics of medicinal substances. For example, bevacizumab, produced by RPE cells, is released via CD63+ vesicles [120]. The scientific literature offers extensive evidence supporting the beneficial effects of stem cell-derived EVs in various diseases. In a model of laser-induced retinal injury, MSC-derived EVs protected RGCs by reducing MCP-1 levels, preventing inflammation and apoptosis [62]. MSC types rich in PEDF and VEGF-AA, growth factors that shield RGCs from neurodegeneration, release more exosomes when exposed to TNF-α [121]. Argonaute-2 knockdown abolished the biological impact of MSC-EVs, suggesting that their miRNA content plays a crucial role in RGC protection in certain disease models [63]. The majority of research on MSC-derived EVs has focused on their benefits in treating diabetic retinopathy (DR) or hyperglycemia damage [122]. Recent studies have improved our understanding of MSC-EV absorption and distribution in the retina through in vitro, ex vivo, and in vivo models [123]. MSC-derived exosomes from bone marrow have been shown to suppress GFAP, VEGF, NF-κB, and iNOS while also altering the Wnt/β-catenin pathway in streptozotocin-induced diabetic rats. Additionally, MSC exosomes can prevent the retinal vascular endothelium from expressing VEGF-A [124]. In streptozotocin-induced diabetic rabbits, retinal degeneration was reduced after MSC-exosome injection, likely due to the activation of miR-222 [125]. Another study identified miR-192 as the cargo within MSC-EVs and hypothesized that it might slow the progression of DR by negatively regulating the integrin-induced subunit α1 (ITGA1). Other miRNAs, such as miR-126, have also been shown to protect against hyperglycemia-induced retinal degeneration by decreasing the effects of HMGB1 and suppressing the activities of NLRP3 and NF-κB [126,127,128]. Additionally, it has been demonstrated that delivering miR-17-3p and miR-18b via exosomes from human umbilical cord MSCs lowers inflammation and apoptosis in DR models based on mice and rats [129]. While EVs are generally considered less immunogenic than cell therapy, the potential for immunological responses, particularly in inflamed or diseased retinal settings, remains a concern. Exosomes may avoid immune recognition, but this has not been fully explored for long-term therapeutic use. Therefore, further investigation of the immunological effects of EVs and exosomes is essential to ensure their safety and effectiveness in treating retinal diseases [127,128]. Table 2 outlines the studies on EV-based treatments for ocular disorders in chronological sequence.

### 3.1. Therapeutic Pathways of EVs for Retinal Regeneration

In retinal degenerative diseases, stem cell-derived EVs can effectively promote tissue repair, cell protection, and other functions comparable to or even surpassing the cell-replacement effects of stem cells while exhibiting a lower tumorigenicity rate [130]. However, further research is necessary to fully elucidate the therapeutic mechanisms of exosomes, particularly the specific components and pathways involved in their protective effects for retinal degeneration (RD). Figure 3 illustrates the therapeutic pathways for retinal regeneration via extracellular vesicles.

#### 3.1.1. Phosphoinositide 3-Kinase (PI3k/Akt) Signaling

The serine/threonine kinase Akt, or Protein Kinase B, serves as a central hub for downstream signaling components, including cytokines, growth factors, and other ligands that stimulate cell development. Activated by a variety of stimuli, the classical PI3K/Akt pathway mediates cellular functions such as survival and proliferation. MSC-derived exosomes (MSC-Exos) can also influence target cell activity through this mechanism [131]. Inflammation and oxidative stress play significant roles in the development of retinal degeneration. Retinal cells are particularly vulnerable to damage from reactive oxygen species. MSC-Exos contains five enzymes essential for producing adenosine triphosphate (ATP) during the glycolysis process. CD73 on the surface of MSC-Exos can increase ATP production and act as an antioxidant by activating Akt and glycogen synthase kinase 3. The toll-like receptor 4 (TLR4), a key target of cellular metabolism, was initially thought to be involved solely in innate immunity. However, research has revealed that TLR4 is expressed in MSCs and plays a protective role in injured nerve cells [132]. By stimulating the TLR4/PI3K/Akt signaling pathway, MSC-EVs can influence MSC migration, growth, and differentiation in addition to reducing cellular inflammation. Co-culturing ESCs with RPE cells can promote RPE cell proliferation and slow down the aging process in vitro by activating the PI3K pathway. ESC-EVs may potentially promote the counter-differentiation of Müller cells into retinal precursors by inhibiting the synthesis of octamer binding transcription factor 4, which is involved in retinal cell renewal [133]. ESCs can downregulate key effectors of cell senescence, such as p53, p21 (WAF1/CIP1), and p16 (INK4a), and upregulate cyclins, such as cyclins A2, B1, and D1. These signaling pathways, mediated by transforming growth factor-β (TGF)-β and PI3K signaling factors, are activated through TGFb1, SMAD3, ID1, ID3, PIK3CG, PDK1, and PLK1. This slows down the aging process of RPE cells by encouraging cell division and preventing cell cycle arrest caused by cyclin-dependent kinase inhibitors, such as p21 [133].

#### 3.1.2. TGF-β1

TGF-β1, a member of the TGF-β superfamily, is associated with cell death and plays diverse roles in biological pathways, including immunosuppression. The molecules of the TGF-β superfamily, Smad protein, can transduce signals into the cell nucleus and influence transcriptional control. TGF-β1 can directly activate Smad2, which, when phosphorylated, interacts with activated TGF-β1 to mediate cell life functions. MSC-derived exosomes can improve liver fibrosis by inhibiting Smad2 phosphorylation, reversing the liver epithelial/mesenchymal transition, and lowering TGF-β1 expression [134]. When internalized by tubular cells and macrophages, MSC-EVs can reverse chronic inflammation. EV-enriched molecules like IL-4 and IL-10 can drive the transition from proinflammatory (M1) to trophic (M2) macrophages. Additionally, EVs can reduce various proinflammatory cytokines, including MCP-1, IL-1β, TNF-α, and IL-6 [134]. Similarly, EVs released by the RPE can prevent T cell proliferation in the presence and absence of inflammation-promoting cytokines such as TN1F-α, IFN-c, and IL-1β. EVs often exhibit immunomodulatory actions, such as upregulating TGF-β1, producing an immunoregulatory phenotype monocyte (particularly the classical phenotype), or reducing the number of monocytes, to mitigate potentially harmful inflammatory responses [132].

#### 3.1.3. MCP-1

The outer nuclear layer (ONL) is composed of the nuclei of PRs. Damage to this structure commonly occurs in retinal degeneration (RD) and can lead to structural abnormalities and alterations in the thickness of the inner nuclear layer and ONL in associated retinal areas [135]. Lipopolysaccharide, a bacterial endotoxin, is known to exacerbate the expression of miR-21, MCP-1, and IL-6, impairing RPE cells. Recent studies suggest that MSC-Exos can confer retinal protection by downregulating the messenger RNA expressions of MCP-1, TNF-α, and ICAM1, thereby reducing retinal inflammation, preventing the accumulation of inflammatory cells like macrophages, improving the structure of the ONL, and safeguarding photoreceptors [132].

#### 3.1.4. Wingless/Integrated Wnt

Wnt signaling pathways, particularly the Wnt/β-catenin system, are involved in embryonic development and tumor formation, primarily through the intracellular or paracrine transmission of signals. Proteins involved in Wnt signaling pathways are often overexpressed in cancer cells, promoting active cell development [136]. In retinal degeneration, activating a Wnt signaling pathway in the RPE can contribute to retinal self-defense. This pathway may promote retinal cell regeneration and prevent photoreceptor degradation [137]. Wnt within MSC-Exos mediates cell activity by triggering an autocrine or paracrine Wnt pathway and acting on the target cells. Additionally, Wnt/β-catenin signaling has been shown to contribute to the development and maintenance of human iPSC-RPE [132].

#### 3.1.5. microRNAs

MicroRNAs (miRNAs) are single-stranded, non-coding RNAs that regulate gene expression. Various miRNAs can be found within EVs. The type and quantities of miRNAs present in exosomes and EVs derived from retinal RPE are closely correlated with retinal tissue inflammation and the prevalence of degenerative diseases. The miRNAs contained within exosomes derived from stem cells can bind to specific genes, thereby regulating the expression of relevant genes and the functions of targeted cells [132].

**Table 2 ijms-26-04078-t002:** Chronological List of Studies on EV-Based Treatments for Ocular Disorders.

Type of Study	Model	Findings	References
Retinal ischemia			
In vivo	C57BL/J mice model of oxygen-induced retinopathy	When administered intravitreally, MSC exosomes lessen the degree of retinal ischemia.	[138]
In vitro/in vivo	Oxygen/glucose deprivation in R28 cells, Wistar rats as a model of retinal ischemia	Exosomes produced from MSCs, when injected intravitreally, enhance retinal healing and reduce the intensity of neuroinflammation.	[139]
In vivo	Rat retinal detachment model	Exosomes derived from MSCs prevent PRs from dying and preserve the integrity of the retina.	[140]
In vivo	Rat model of retinal ischemia	Retinal microglia inflammation is reduced by engineered EVs overexpressing miR-424, which also lessens the impact of ROS on Müller cells and microvascular endothelial cells.	[139]
Optic nerve crush/injury			
In vivo	Rat model of ONC	The administration of exosomes produced from umbilical MSCs enhances glial cell activation and RGC survival.	[141]
In vitro/in vivo	RGCs, rat model of ONC	RGCs are protected against neurodegeneration by AAV2 expressing the six miRNAs found in exosomes from bone marrow MSCs.	[142]
In vivo	Rat model of ONC	RGC regeneration is aided by EVs derived from human MSCs.	[143]
In vivo	Rat model of ONC	RGC regeneration is induced by intravitreal injection of rat MSC exosomes.	[129]
Corneal disorders			
In vitro/in vivo	Human corneal epithelial cells, mice with corneal debridement	Exosomes derived from human corneal MSCs expedite the healing process of a corneal lesion.	[144]
In vivo	Murine model of a corneal wound	EVs derived from corneal stromal stem cells involve miRNAs for regenerative.	[145]
In vivo	Corneal epithelial defect model based on rats	Compared to exosomes isolated from MSCs, those from iPSCs elicit greater therapeutic outcome.	[146]
In vitro	Human corneal endothelial cells cultured with serum deprivation and tunicamycin	EVs isolated from MSCs protect human corneal endothelial cells from the apoptosis induced by ER stress.	[147]
In vivo	In vivo model of a corneal scar	Exosomes produced from MSCs and iPSCs may be placed onto a thermosensitive hydrogel to facilitate corneal regeneration.	[148]
In vitro	Primary corneal stromal cells	Human corneal stromal cells are encouraged to migrate and regenerate by secretory exosomes.	[149]
In vitro/in vivo	Human corneal epithelial cells, rats	Through the release of miR-21, EVs derived from human umbilical MSCs aid in the healing of a corneal lesion.	[150]
In vitro	Corneal endothelial cells	EVs made from platelets cause the corneal endothelium to heal.	[151]
In vivo	Mice with corneal injury	Repair of a corneal wound is accelerated by exosomes produced from MSCs that are treated with siRNAs against c-Rel.	[152]
Other ocular disorders			
In vivo	Patients with refractory macular holes	Exosomes produced from MSCs aid in the healing process following macular hole repair.	[153]
In vitro/in vivo	RPE cells, laser retinal injury mouse model	MSC exosomes had positive effects on retinal injuries and blue light-stimulated RPE cells.	[154]
In vitro	Human lens epithelial cells	Exosomes produced from adipose stem cells shield HLECs from UVB oxidative damage.	[155]
In vitro/in vivo	HEK-293 T, C57BL/J mice	Administration of AAV linked with exosomes facilitates improved delivery of the *RS1* gene potential effects on X-linked retinoschisis therapy.	[130]
In vitro/in vivo	Human skin fibroblasts, ARPE-19, C57BL/J mice model of retinal fibrosis	Through EMT suppression, miR-27b in exosomes produced from human umbilical cord MSCs prevents retinal fibrosis.	[156]
In vitro/in vivo	T-cells, Lewis rats as a model of experimental autoimmune uveitis	Exosomes from immunized uveitis rats prevent T cells from mounting an immunological response.	[157]
In vitro/in vivo	PC12 cells, rat model of retinal degeneration	Retinal degeneration is reduced by EVs derived from dental stem cells.	[158]
In vivo	C57BL/J and NCG mice, patients with chronic GVHD-associated dry eye	Through the activity of miR-204, exosomes from MSCs given as eye drops alleviate the symptoms of GVHD-associated dry eye in humans and rats.	[159]
In vivo	Laser-induced choroidal neovascularization mouse model	Mouse retina-derived engineered exosomes have positive impacts on choroidal neovascularization.	[160]
In vivo	Mouse model of dry eye disease	Ascorbic acid-associated MSC-derived exosomes promote corneal epithelial healing, decrease inflammation, and increase tear production.	[150]
In vitro	3T3-L1 preadipocytes and mouse retina microvascular endothelial cells	Microvascular endothelial cells become dysfunctional when exposed to high glucose, which is triggered by LINC00968 in exosomes obtained from 3T3-L1 cells.	[161]
Ex vivo/in vivo	Vitreous from post-mortem donors or C57BL/J mice	Use of vitreous liquid biopsies because it is rich in exosomes that carry retinal proteins.	[162]

RPCs: Retinal Progenitor Cells, NR: Neural Retina, CNS: Central Nervous System, RSCs: Retinal Stem Cells, PRs: Photoreceptors, CMZ: Ciliary Marginal Zone, CB: Ciliary Body, RPE: Retinal Pigment Epithelium, IPE: Iris Pigment Epithelium, ESCs: Embryonic Stem Cells, hESC: Human Embryonic Stem Cells, iPSCs: Induced Pluripotent Stem Cells, hiPSCs: Human Induced Pluripotent Stem Cells, RGCs: Retinal Ganglion Cells, NSCs: Neural Stem Cells, NPCs: Neural Progenitor Cells, AHPCs: Adult Hippocampal Progenitor Cells, MSCs: Mesenchymal Stem Cells, BMSCs: Bone Marrow Stem Cells, ROS: Reactive Oxygen Species, ONL: outer Nuclear Layer, AAV: Adino Associated Viruses, ER: Endoplasmic reticulum, HLEC: Human Lymphatic Endothelial Cells, UVB: Ultra violet Light-B, EMT: Epithelial to Mesenchymal Transition.

It has been reported that exosomes derived from umbilical cord MSCs can have a neuroprotective effect on neurons in rats with high blood sugar [163]. Beyond miRNAs, EVs and exosomes carry other bioactive molecules that support retinal protection. For example, the lncRNA, SNHG7, in an in vitro model of human retinal microvascular endothelial cells treated with high glucose, was able to prevent the epithelial-to-mesenchymal transition by reducing the activity of miR-34a-5p [164]. MSC-derived EVs have shown positive effects on glaucoma and retinal degeneration in both in vivo and in vitro models of AMD and glaucoma [165,166]. MiR-21 is a key contributor to the neuroprotective effects observed in retinal degeneration [167]. EVs generated from embryonic stem cells can also halt retinal degeneration by upregulating Oct4 [166]. In an in vivo model of retinitis pigmentosa, other miRNAs, such as miR-146aNr4a3, demonstrated comparable anti-inflammatory responses in rd10 mice [168]. While MSC-derived EVs are beneficial in various retinal pathologies, it is important to note that in vitro experiments can be context-dependent. The response elicited by these vesicles can be influenced by the conditions under which the cells were cultured. For instance, EVs from cells cultured in high-glucose/hypoxia conditions have been shown to induce angiogenesis and a DR-like phenotype [169]. Exosomes derived from adipocytes produced similar outcomes [168]. Additionally, exosomes generated from MSCs in a hypoxic environment, whether administered intravitreally or subretinal, were unable to enhance a model of hereditary retinal degeneration, even in the presence of CD34+ cells [170]. Despite the abundance of data demonstrating the favorable influence of RPE-derived exosomes on other cell types, to our knowledge, there is only one published study describing their potential therapeutic implications for recovering PRs homeostasis. This study was conducted using an N-methyl-N-nitrosourea-induced retinal degeneration model in mice [171]. It is worth noting that in vitro studies have shown that environmental stressors can impact the cargo carried by exosomes, potentially leading to harmful signals [172]. Limited data is available on neural and retinal progenitor cells, but existing studies support the beneficial effects of exosomes derived from these cells in preventing retinal degeneration in animal models [162,173].

### 3.2. Advantages, Limitations, and Future Perspective of EV-Based Therapies

EVs offer promising potential for treating and diagnosing various ocular diseases, as evidenced by the literature. However, advancements in current technology are necessary to authorize new EV-based therapeutic approaches. Currently, cell-based therapy and AAV2-based gene therapy for monogenic disorders are the two most extensively researched methods for treating various diseases. Despite favorable results, both strategies are classified as “Advanced Therapy Medicinal Products” (ATMPs), subject to specific regulations by the FDA and EMA. This can lead to varying regulatory requirements and marketing recommendations in different countries. For instance, the two regulating bodies now have different opinions for ATMPs [174]. Compared to cells, EVs offer several advantages that may circumvent some limitations associated with cell-based therapy and facilitate the development of novel EV-based medicines. EVs have a more stable protein payload and are easier to store than cells [175,176].EVs are suitable for a wider range of applications due to their less complex intravenous injection and biodistribution compared to cells [174]. Environmental influences do not affect their modularized component [177]. There is no chance of a damaging immune response or malignant change [178].

The International Society for Extracellular Vesicles (ISEV) has emphasized that EVs should be regulated based on their functional moiety. Any EV-based therapy must be classified as a specific medical product to comply with corresponding regulations [179]. ATMP recommendations may need to be considered for subgroups of EVs with significantly altered cellular origins [180]. At present, there are several obstacles hurdles with using EVs as a biological therapy that still need to be resolved. There are over 190 isolation techniques, and no standardized guidelines exist [181]. The storage and isolation processes have the potential to degrade the overall quality of the sample, which introduces biases and variables that may make it more difficult to compare research conducted in various labs. To be more precise, storage circumstances can change the phenotype and stability of EVs, possibly causing material loss. Moreover, no thorough comparison of the quality of “fresh” and “stored” EVs has ever been made, as far as we are aware. To standardize the usage of these vesicles and make results from other laboratories comparable, the source from which the EVs were obtained should also be precisely described. In vitro studies should specify the exact FDA-approved dosage due to in vivo instability [182]. Because EVs are stable in all body fluids, they offer a useful alternative to cell-penetrating peptides (CPPs). EVs carry various molecules (such as proteins and short RNAs) from a donor to a destination to deliver information about the vesicle and particular signals to nearby cells. Furthermore, in cases of retinal illnesses, EVs may be employed as indicators of therapy response as well as disease severity and progression. Hernandez et al. [183] demonstrated that exosomes can be released by polarized primary RPE cells in response to extended low-level oxidative stress; additionally, the protein signature of these vesicles was different from that of normal RPE cells and was noted prior to the emergence of morphological changes and cell dysfunction. Based on these findings, EVs might be used as potential pre-symptomatic markers for eye conditions. Interestingly, iPSC-RPE cell lines derived from individuals at high risk for AMD showed the same protein profile in the absence of oxidative stress, underscoring the potential of EVs for both diagnosis and prognosis.

EVs can be used to deliver miRNAs linked to the regulation of autophagy in AMD [184]. Two primary administration routes may be used with this strategy and others based on various therapeutic molecules (such as proteins, oligonucleotides, agomiRs, and antagomiRs). One method involves isolating EVs from donor cells and incorporating the therapeutic material in a cell-free environment; another technique involves treating donor cells with the relevant molecules in vitro, allowing them to internalize the molecules, and then isolating the enriched EVs [142]. Depending on the molecules being employed, the latter strategy may need significant implementation. Utilizing the direct pathway of the vitreous humor and the phagocytic activity of RPE cells, vesicles made using both of these techniques may be injected intravitreally or subconjunctivally [142]. Compared to the previously discussed methods, topical administration would be a more beneficial and secure substitute. It might be advantageous to generate EVs using cells from ocular tissues. This strategy could be a useful substitute to address problems with EV separation from the eye or its fluids. Even though this source can identify biomarkers linked with sickness, little is known about the extraction of vesicles from tears [185]. The need for better separation techniques, the availability of free proteins in these biological materials, and the necessity of repeated sample collection to achieve sufficient volume for EV isolation are further restrictions on the use of tears, aqueous humor, and vitreous bodies in treatment [147]. According to reports [186,187,188,189], blood is a potentially simple specimen to obtain and separate EVs for molecular characterization and potential therapeutic uses. EVs produced from low-risk healthy persons may be beneficial for high-risk patients or patients who are already afflicted with a disease. Based on the positive data compiled in this study, we believe that EV-based treatments might be a valuable therapy option for many ocular conditions. The potential benefits of EV-based therapies extend beyond the retina and associated conditions, which have been extensively covered in this study. Other ocular tissues, such as the cornea and vitreous, have also shown promising outcomes in a range of ocular illnesses. Beneficial substances (such as proteins, lipids, nucleic acids, or synthetic therapeutic molecules) may be carried by EVs and utilized as protective agents for patients with non-advanced stages of illnesses or as a preventative measure for high-risk patients. EVs appear to be more appropriate for ophthalmological applications when compared to alternative strategies, such as those based on nanoparticles (NPs) (e.g., nanoemulsions, nano micelles, quantum dots, liposomes, or inorganic NPs). Even though pre-clinical data on NPs appear promising, long-term toxicological evaluations are still lacking, and short-term toxicity has occasionally been found (such as subacute toxicity of rabbit vitreous and sub-conjunctival cavities) [190]. The potential use of EV-based treatments in multifactorial ocular illnesses (such as AMD, glaucoma, and DR), which are brought on by a complex interplay between hereditary and environmental variables, is additional advantage. With certain limitations, intriguing gene-based treatments have been discovered for monogenic illnesses like RP. Specifically, a phase III clinical trial including 31 patients showed promising outcomes for the gene therapy licensed for RP, Luxturna^®^ (voretigene neparvovec). However, this therapy can benefit only subjects carrying mutations in the *RPE65* gene, detectable in a small sub-population accounting for 0.3–1% of all RP patients [191]. It is well known that mutations in several genes are linked to inherited retinal degeneration (IRD) syndromes. According to the Retino Genetics database (http://www.retinogenetics.org/), 186 different genes have been linked to IRDs [191].

Given the array of IRD-causing mutations, developing gene therapies to address each specific mutation presents significant challenges in terms of feasibility, cost, and timeline. EV-based therapies, tailored to specific pathologies or shared target features of different IRD syndromes, offer a potential approach to prevent or slow down retinal degeneration in the early stages of various disorders. These therapies can be used alone or in combination with other treatments to overcome limitations. For ocular disorders, several therapeutic approaches have been proposed, each with varying degrees of constraints. AAV vector-based therapies are invasive, CRISPR/Cas gene editing yields suboptimal results, siRNAs degrade easily, and ethical concerns regarding iPSCs and ESCs, including epigenetic memory and teratogenicity, hinder their use [191]. Optogenetics, which has shown positive results in a blind patient, offers a potential alternative to gene therapy, resulting in partial restoration of visual function [192]. Cost is a crucial factor to consider for such therapy. Gene therapies can be expensive to develop and produce, with costing up to USD 3 million, limiting their widespread clinical use [193]. EV-based methods, being less expensive, offer a more attractive option for broader clinical applications. Table 3 provides a chronological list of studies exploring the role of EVs in retinal pathology, proliferative vitreoretinopathy (PVR), and RP. Figure 4 illustrates the diverse roles played by exosomes derived from stem cells in addressing various eye diseases.

## 4. Small Molecules Used for Inducing Retinal Cell Differentiation

Retinal cell differentiation is a crucial area of research in regenerative medicine, seeking innovative treatment strategies for retinal diseases such as retinitis pigmentosa, AMD, and DR. Small molecules, capable of modifying specific signaling pathways and biological processes, have emerged as promising agents for inducing retinal cell differentiation [207,208,209]. Table 4 lists the small molecules that are used to induce differentiation of the retinal cells.

### 4.1. Retinoic Acid

Retinoic acid (RA), a derivative of vitamin A, is one of the most extensively studied small molecules for promoting retinal cell development. RA stimulates the differentiation of retinal progenitor cells (RPCs) into photoreceptors, a critical process in eye development. RA therapy increases the expression of genes specific to photoreceptors in RPCs, suggesting its potential for treating photoreceptor regeneration [210].

### 4.2. Sonic Hedgehog Agonists

The retina relies on the Sonic Hedgehog (Shh) signaling system for growth and maintenance. Purmorphamine, an Shh pathway antagonist, has been shown to promote RPC differentiation and proliferation. Shh agonist therapy increased the number of retinal ganglion cells (RGCs) in vitro, suggesting its potential for RGC replacement treatments [211].

### 4.3. Histone Deacetylase Inhibitors

Valproic acid (VPA) and other histone deacetylase inhibitors (HDACi) have been used to modify gene expression and chromatin structure, thereby stimulating retinal cell differentiation. VPA has been shown to facilitate the development of iPSCs into retinal cells. VPA therapy increases the expression of RGC markers in retinal cells derived from iPSCs, suggesting its potential for retinal regeneration [212].

### 4.4. Fibroblast Growth Factor (FGF)

FGFs family, particularly FGF2, support the proliferation and differentiation of retinal progenitor cells. FGF2 can facilitate the development of hESCs into retinal neurons, such as photoreceptors and RGCs, suggesting its potential for retinal cell replacement therapies [212].

### 4.5. Wnt Signaling Modulators

The Wnt signaling inhibitor IWR-1 has been shown to stimulate the differentiation of retinal progenitor cells into photoreceptors. IWR-1 successfully stimulates the expression of photoreceptor markers in retinal progenitor cells, suggesting its potential for use in treating retinal degenerative diseases [213]

### 4.6. Y-27632

The ROCK inhibitor Y-27632 has been shown to enhance the survival and development of retinal progenitor cells. Y-27632 stimulates the development of human iPSCs into RPE cells, which are crucial for maintaining photoreceptor survival and function [214].

### 4.7. SB431542

The TGF-β pathway inhibitor SB431542 has been shown to promote the development of retinal cells derived from stem cell sources. SB431542 facilitates the differentiation of human iPSCs into retinal neurons, suggesting its potential applications in regenerative medicine [215].

**Table 4 ijms-26-04078-t004:** Small Molecules Used for Inducing Retinal Cell Differentiation.

Small Molecule	Target Pathway	Cell Source	Model	Effect	Principal Findings	Reference
Retinoic Acid (RA)	Retinoid Signaling	Retinal Progenitor Cells (RPCs)	In vivo	Encourages the growth and development of photoreceptors	Increases the expression of genes unique to photoreceptors, indicating promise for photoreceptor regeneration.	[216]
Purmorphamine	Sonic Hedgehog (Shh)	RPC	In vitro	Increases the quantity of RGCs in the retina	Encourages RPC growth and maturation.	[217]
Valproic Acid (VPA)	Histone Deacetylase (HDAC)	iPSCs	In vitro	Increases RGC marker expression	Allows iPSCs to differentiate into RGCs.	[218]
FGF2	Fibroblast Growth Factor	hESCs	In vivo	Helps retinal neurons, such as photoreceptors and RGCs to mature	Encourages the growth and differentiation of hESCs into retinal cell types.	[219]
CHIR99021	Wnt Signaling	hESCs	In vitro	Encourages the development of RGCs	Alters Wnt signaling to improve hESC differentiation into RGCs.	[220]
IWR-1	Wnt Signaling Inhibition	RPCs	In vitro	Increases photoreceptor marker expression	RPC differentiation into photoreceptors by Wnt signaling.	[213]
Y-27632	ROCK Inhibition	iPSCs	In vitro	Improves the ability to survive and differentiate into RPE cells	Boosts iPSC differentiation into RPE cells and increases survival.	[221]
SB431542	TGF-β Pathway Inhibition	iPSCs	In vitro	Makes it easier for retinal neurons to differentiate	Prevents TGF-β signaling to help iPSCs differentiate into retinal neurons.	[222]
DAPT	Notch Signaling Inhibition	RPCs	In vitro	Encourages the differentiation of photoreceptors	RPC development into photoreceptors is enhanced.	[223]
Taurine	Amino Acid Supplementation	RPCs	In vivo (Zebrafish model)	Improves photoreceptor survival and differentiation	Taurine supplementation improves photoreceptor survival and differentiation.	[224]
Forskolin	Adenylate Cyclase Activator	RPCs	In vitro	Encourages the differentiation of the retinal cells	Promotes the development and survival of retinal progenitor cells by activating adenylate cyclase.	[225]
Thiazovivin	ROCK Inhibition	Human Pluripotent Stem Cells (hPSCs)	In vitro	Improves the differentiation and survival of retinal cells	Enhances survival and differentiation of hPSCs into retinal cells.	[226]
BDNF	TrkB Receptor Activation	RPCs	In vivo	Encourages RGC differentiation and survival	RGCs are more likely to survive and differentiate when exposed to brain-derived neurotrophic factor (BDNF).	[227]
PD98059	MEK Inhibition	RPCs	In vitro	Encourages the development of photoreceptors	Promotes the development of retinal progenitor cells into photoreceptors by inhibiting MEK signaling.	[228]

## 5. Gene Therapy for Inherited Eye Diseases

Gene therapy is a therapeutic approach that involves altering the expression of specific genes in a patient’s cells by introducing genetic material (DNA or RNA). Since the first gene therapy experiment in 1990, treating two infants with severe combined immunodeficiency (SCID) due to ADA deficiency, gene therapy has been explored and used for various inherited and acquired diseases [212]. Over 20 gene therapies have been officially approved for clinical use by various drug regulatory agencies worldwide [229].

### 5.1. The Eye Is an Ideal Target for Gene Therapy

The human eye has long been a prime target for gene therapy due to several factors. First, its immune-privileged status reduces the risk of inflammatory reactions to foreign molecules introduced into the retina. This is primarily attributed to the blood/retinal barrier (BRB), formed by tight junctions between the endothelial cells of the retinal microvasculature and the RPE [229]. Second, the eye’s small, confined space minimizes the required medication dosage and limits the potential for systemic spread of locally delivered vectors. Third, the paired nature of the eyes provides a control group for evaluating treatment safety and efficacy (one needs to keep in mind, however, that the two eyes may not be completely independent and that treatment in one eye may result in an immune or other reaction in the fellow eye). Finally, the eye’s accessibility allows for non-invasive monitoring of treated retinal structure and function using both surgical and diagnostic techniques [230]. Table 5 outlines hereditary retinal dystrophies and associated diseases that may benefit from gene therapy. The current ongoing clinical trials are highlighted in Table 6.

#### 5.1.1. Gene Delivery Systems

It is crucial to differentiate between in vivo and ex vivo gene therapy. In vivo techniques involve directly delivering a gene therapy vector to a living organism, while ex vivo methods entail isolating patient cells, modifying them, and reintroducing them [231]. Although there have been preclinical attempts at gene-corrected cell transplants, ocular gene therapy is an in vivo procedure that involves injecting genetic material directly into the patient’s eye via subretinal or intravitreal injection [232]. A key distinction lies in how nucleic acids are incorporated into target cells. Due to their large size and negative charge, DNA and RNA cannot passively cross cell membranes. To overcome this, various gene delivery methods are employed, primarily categorized as viral and non-viral systems [215].

#### 5.1.2. Viral Delivery Systems

Viruses are the most often used vectors; their method of transduction allows them to enter target cells and release their genetic material. To deliver therapeutic nucleic acids, several recombinant viruses can be used, each having a different cargo limit, integration capacity, transduction efficiency, cellular tropism, and immune response risk. Ads, also known as adenoviruses, is a family of DNA viruses that may infect quiescent or dividing cells and replicate in the host nucleus without integrating into the genome. Because of their huge cargo capacity (between 8 and 36 kb) and versatility in the transduction of several cell types, adenoviruses have been extensively studied as vectors for gene therapy. However, when it comes to IRDs, traditional adenoviral vectors (AVs), which were created by replacing the E1 region with the desired transgene cassette [216], did not work well because some viral genes were expressed in the infected target cells, which increased immunogenicity and reduced treatment survival even in the immunologically favorable human eye [233]. These issues have been mostly handled with second- and third-generation vectors, which eventually eliminate all viral coding sequences and include helper-dependent AVs. The human genome has more than 20 integration sites for defective single-strand (ss) DNA parvoviruses or adeno-associated viruses (AAVs). Replication-competent AVs, vector instability, and infecting helper viruses have all been reported with problems [234]. Recombinant AAVs (rAAVs) have a favorable immunologic profile, can provide stable transgene expression, have extended retinal tropism, and are non-integrating in the absence of rep protein (which lowers the risk of insertional mutagenesis, unlike (lentiviruses) LVs). Unlike AVs, rAAVs do not carry any virus open reading frame. rAAVs are the most widely utilized vector in gene therapy techniques for IRDs because of these reasons. As of right now, primates have been used to identify 13 naturally occurring AAV serotypes (AAV1–AAV13). Each serotype differs in terms of its capsid shape and other characteristics, particularly tropism [234]. Furthermore, AAVs can be altered in several ways. For instance, one method, known as pseudotyping or cross-packaging, involves enclosing the viral genome containing the transgene into the capsid of an AAV serotype that is distinct from the original (for example, an AAV2/8 vector is a pseudo type in which the genome of an AAV2 serotype is packaged into an AAV8 capsid) [234]. Selecting the right serotype and pseudotype is crucial for optimizing vector design for the intended illness. AAV2/5, AAV2/8, and AAV8 are the serotypes and pseudotypes that have been employed thus far in IRD clinical studies. The primary drawback of rAAV vectors is their payload capacity, which is restricted to 4.7 kb. Although with a reduced photoreceptor transduction efficiency, dual AAV vectors, each of which contains half of a large transgene expression cassette, have been shown to improve retinal phenotype in murine models of IRDs [235,236]. For certain IRDs whose causal gene coding sequence exceeds the 4.7 kb limit, such as ABCA4-related Stargardt’s disease and MYO7A-related Usher’s syndrome type 1B, lentiviruses (LVs) are retroviruses with a greater packing capacity (8 kb), which makes them an excellent option to AAV vectors. Currently under investigation for IRDs include the equine infectious anemia virus (EIAV) and the retroviral variation of human immunodeficiency virus type 1 (HIV-1). There are two basic problems with LVs. First, genomic integration occurs naturally in LVs [237]. While this can result in the long-term expression of foreign DNA, it also entails the danger of insertional mutagenesis [238]. Given that LVs episomes may also produce stable transduction in post-mitotic tissues like the retina, such a risk may not be warranted in the case of IRDs. Integration-deficient lentiviral vectors (IDLVs), which have been effectively applied in a mouse model of retinal degeneration, can be utilized to get around this restriction. Second, LVs are capable of effectively transducing RPE cells and, only to a lesser extent, which is generally insufficient for therapeutic purposes, differentiated photoreceptors [239].

#### 5.1.3. Non-Viral Delivery Systems

Non-viral delivery methods offer several advantages over viral systems, including the potential for unlimited cargo capacity, simultaneous transport of multiple therapies, minimal immunogenicity, and cost-effective production. These methods employ physicochemical agents to condense DNA and/or facilitate its passage across cell membranes [237]. However, physical techniques like sonoporation and electroporation (which use ultrasound or electricity to temporarily increase cell permeability) and direct DNA injections are susceptible to enzymatic degradation of therapeutic nucleic acids. Chemical agents such as lipopolyplexes, cationic liposomes, and nanoparticles can protect the payload from nuclease activity [237]. Although non-viral DNA systems have shown promise in preclinical gene therapy studies, their transient gene expression often limits their applicability in vivo, leading to generally ineffective delivery [240].

### 5.2. Gene Therapy Approaches for Retinal Diseases

Retinal gene therapy approaches vary based on specific mutation, employing techniques such as gene replacement, augmentation, editing, silencing, or modifier therapy. Gene replacement or augmentation involves introducing a functional copy of a defective gene to increase the production of the correct protein. Gene editing corrects gene mutations or reduces mutant protein production to address disease conditions. Gene silencing, applicable to acquired disorders, uses RNA interference (RNAi) to suppress the abnormal expression of the targeted pathogenic protein. Modifier gene therapy targets genes that can influence upstream or downstream pathways of the defective gene, altering their expression to improve cellular function [235].

#### 5.2.1. Gene Replacement Therapy

Gene replacement is a straightforward approach that addresses gene deficiency by introducing a functional copy of a damaged or nonfunctional gene. This method does not alter the defective gene but rather complements its missing function. Monogenic recessive hereditary disorders are prime candidates for gene replacement therapy. For example, mutations in the *CEP290* gene account for 15–20% of Leber congenital amaurosis (LCA) cases [237]. Patients with LCA due to *RPE65* mutations (5–10%) can benefit from Luxturna (Spark Therapeutics, Inc., Philadelphia, PA, USA), which delivers a functional *RPE65* gene. Despite Luxturna’s clinical success, its efficacy and durability have been questioned in patients with diverse genetic backgrounds, and limitations include the need for residual tissue before treatment and challenges in measuring treatment outcomes [241]. Retinal disorders can be caused by various gene mutations with different inheritance patterns, including autosomal dominant, autosomal recessive, and X-linked. Retinitis pigmentosa (RP), for example, has been linked to over 3000 mutations in approximately 70 genes, but a genetic cause remains unidentified among many RP patients. Furthermore, gene replacement is not suitable for all gene mutations, including dominant mutations, large genes that cannot be packaged in current delivery vectors, and polygenic conditions. The cost of individual gene replacement therapies for each mutation would limit patient access [242]

#### 5.2.2. Gene Silencing

Age-related macular degeneration (AMD), glaucoma, and other visual disorders are being addressed using gene silencing techniques that target *vascular endothelial growth factor (VEGF)* with small interfering RNA (siRNA) or microRNA (miRNA). Multiple clinical trials are currently investigating these targeted gene silencing approaches. However, challenges such as RNA instability, low bioavailability, and nonspecific targeting leading to off-target effects have hindered the advancement of clinical trials in the ocular field beyond phase III [243].

#### 5.2.3. Gene Editing

Gene editing modifies disease conditions by correcting gene mutations or reducing mutant protein production. Various gene editing methods have been developed, including homing endonucleases, meganucleases, zinc finger nucleases (ZFNs), transcription activator-like effector nucleases (TALENs), and CRISPR/Cas9. Among these, the CRISPR/Cas9 system is the most widely recognized and holds significant promise in gene therapy [237]. The CRISPR/Cas9 system consists of a guide RNA specific to the target gene and an endonuclease that creates site-specific double-stranded DNA breaks, enabling precise and permanent genetic modifications [244,245]. While CRISPR can address single-gene disorders, it is less effective in individuals without a confirmed genetic diagnosis. EDIT-101 (Editas Medicine, NCT03872479) is a gene therapy for LCA type 10 that targets the *IVS26* mutation, which causes incorrect splicing between exons 26 and 27, leading to a prematurely truncated *CEP290* gene. This therapy uses CRISPR/Cas9 to delete or invert the IVS26 mutation, restoring proper splicing and *CEP290* function [246,247]. Prime editing (PE) is a promising new technology that can potentially repair various mutations, including small insertions and deletions. It utilizes reverse transcriptase and Cas9 to correct genomic mutations. However, several challenges must be addressed for PE to effectively treat genetic diseases in the future, including the large number of genes involved in disease etiology [246,247].

##### Advancements in CRISPR/Cas9 Gene Editing for Retinal Regeneration

Advances in recent times in the application of CRISPR/Cas9 gene editing have brought remarkable progress in retinal regeneration and presented the clinical community with potentially effective therapy for inherited retinal diseases (IRDs) [245,248,249]. The milestone BRILLIANCE trial tested the efficacy and safety of EDIT-101, which is a CRISPR-targeted therapy for the *CEP290* gene mutation for Leber Congenital Amaurosis 10 (LCA10). The research proved that 11 of the 14 patients had measurable improvement in vision, which indicates the promise of in vivo gene editing for retinal disease. These clinical results were complemented by preclinical research into CRISPR/Cas9 use in animal models [250]. Researchers were able to correct a mutation in a mouse model of retinitis pigmentosa, restoring the production of rhodopsin and enhancing retinal function and structure [251]. Moreover, large-scale CRISPR screens have isolated context-specific genetic regulators of retinal ganglion cell regeneration and offered insights into potential therapeutic targets. In addition, retinal organoid development has allowed the evaluation of CRISPR/Cas9 gene editing efficiency and provided a scalable platform to evaluate therapeutic interventions. These cumulative breakthroughs emphasize the revolutionary capability of CRISPR/Cas9 in retinal regeneration, paving the way for novel therapies against IRDs [252,253].

#### 5.2.4. Modifier Gene Therapy

Since the early 1940s, it has been recognized that modifier genes can influence the expression of other genes, especially mutant genes, without affecting healthy phenotypes [237]. This has led to growing interest in developing treatments that leverage modifier gene functions for disease therapy. Modifier genes can influence the severity of a gene mutation’s associated trait. By targeting upstream or downstream pathways of multiple defective genes, modifier gene therapy can address clinical symptoms in a mutation-agnostic manner without requiring a genetic diagnosis. However, identifying the specific modifier genes involved in a particular disease manifestation can be challenging and costly, significantly impacting the development of modifier medicines [237,246]. Research into modifier gene therapy has accelerated for both retinal and systemic disorders, including neuromuscular disease, cystic fibrosis, spinal muscular atrophy, and cancer. Previous studies have demonstrated the ability of modifier genes, such as nuclear hormone receptors, to “reset” various networks associated with retinal disease phenotypes, including phototransduction, metabolism, cone cell development, inflammation, and cell survival, restoring retinal homeostasis. OCU400 (Ocugen, Inc., Malvern, PA, USA) is a nuclear hormone receptor-based modifier gene therapy currently undergoing clinical trials for retinitis pigmentosa (NCT05203939). This gene-agnostic strategy offers several advantages, including reduced research and commercialization costs, by eliminating the need for individual therapies for each mutation. By targeting a broader patient base, modifier gene therapy can address unmet medical needs in the field of rare genetic diseases [237,254]. Figure 5 illustrates gene therapy vectors and techniques commonly used for retinal gene therapy.

**Table 5 ijms-26-04078-t005:** Gene therapy approaches for various inherited retinal dystrophies and related syndromes.

Gene(s) Involved	Clinical Features	Genetic Features	Gene Therapy Approach	Clinical Trials	Therapeutic Mechanism	Outcome/Current Status	References
Usher Syndrome							
*MYO7A*, *USH2A*	AR deafness along with RP. Type 1: Vestibular symptoms, childhood-onset RP, and congenital profound deafness. Type 2: Later-onset RP with congenital partial deafness and absence of vestibular symptoms. Type 3: Adult-onset retrograde hearing loss that progresses from the second to the fourth decade.	*MYO7A* (*USH1B*): 49 exons, 87 kb; opsin trafficking and melanosomes involved. <br> *USH2A* (*USH2*): Splice site mutations resulting in pseudoexon inclusion are present in this 15 kb gene.	*USH2A*: Antisense oligonucleotides (AONs) to bypass exon 13 may be given intravitreally; *MYO7A*: EIAV lentiviral vector (UshStat) supplied by subretinal injection.	*MYO7A*: NCT01505062, NCT02065011 (ongoing) (stopped) *USH2A*: QR-421a (AON), phase 1/2 of STELLAR (NCT03780257), and phase 2/3 trials under planning (SIRIUS and CELESTE).	*MYO7A*: Lentiviral vector-based gene replacement therapy. *USH2A*: Restoring normal mRNA and protein function by exon skipping via AONs.	*MYO7A*: Initial results are encouraging despite a few obstacles. *USH2A*: Preliminary studies demonstrate effectiveness and safety in stabilizing eyesight.	[255]
Choroideremia							
*CHM*	Gradual deterioration of the retina, nyctalopia throughout childhood, restriction of the VF in early adulthood, and eventual legal blindness by the fifth decade. Findings: retinal atrophy, choroid, thinning of the retinal layers, and outer retinal tubulations.	Gradual deterioration of the retina, nyctalopia throughout childhood, restriction of the VF in early adulthood, and eventual legal blindness by the fifth decade. Findings: retinal atrophy, choroid, thinning of the retinal layers, and outer retinal tubulations.	Injecting the AAV2-*REP1* vector subretinally (SRI).	The following are several trials: NCT02671539, NCT02077361, NCT02553135, NCT02341807, and NCT03496012. Trials in Phase III are continuing. IVI in phase I of 4D-110 (NCT04483440).	AAV2 vector-based gene replacement therapy is used to deliver the functional *REP1* gene.	Some individuals vision has stabilized or improved, according to early-phase research. The purpose of ongoing trials is to verify long-term safety and efficacy.	[256,257,258]
X-Linked Retinoschisis (XLRS)							
*RS1*	In 50% of individuals, focal schisis, a spokewheel pattern of folds, and peripheral retinoschisis are present. The first two decades saw a decrease in visual acuity.	Retinochisin, which is involved in retinal cell adhesion, is encoded by RS1. Subunit assembly is disrupted by mutations.	Intravitreal injection of AAV5-mOPs-RS1 and AAV8-scRS/IBPhRS vectors (IVI).	Phase 1/2 trials: NCT02317887 (NEI, AAV8-scRS/IRBPhRS), NCT02416622 (rAAV2tYF-CHhRS1).	Gene replacement therapy using AAV vectors to deliver functional *RS1* gene.	Early studies show mixed effective results but no safety. More research is required to improve expression and delivery.	[259,260,261]
Stargardt Disease (STGD)							
*ABCA4*, *ELOVL4*, *PROM1*	Progressive macular degeneration, retinal flecks, central atrophy surrounded by patchy atrophy. Visual acuity progressively declines.	STGD1 (*ABCA4*): AR, flippase for atRAL/PE. STGD3 (*ELOVL4*): AD, very long-chain fatty acids synthesis. *STGD4* (*PROM1*): AD, plasma membrane organization.	Lentiviral gene therapy (SAR422459, EIAV vector). Non-viral techniques (self-assembled nanoparticles).	SAR422459: Phase I/II (NCT01367444, stopped in 2020). Hybrid AAV dual vector and non-viral delivery in preclinical stages.	Gene replacement therapy using lentiviral vector for *ABCA4*. Exploration of non-viral delivery systems.	Stopped SAR422459 trial due to strategic reasons. Preclinical research is ongoing for alternative delivery methods.	[262,263,264,265]
Achromatopsia (ACHM)							
*CNGA3*, *CNGB3*, *GNAT2*, *PDE6C*, *PDE6H*, *ATF6*	Complete form: No cone function, BCVA < 20/200, no color perception. Incomplete form: Residual cone function, higher VA, some color discrimination. Features: Photophobia, nystagmus, central scotomata, high refractive errors.	*CNGA3* and *CNGB3* (70–80% cases): Encode CNG channel subunits. *GNAT2*: Catalytic G-protein subunit. *PDE6C* and *PDE6H*: Photoreceptor phosphodiesterase subunits. *ATF6*: ER homeostasis, crucial for foveal development.	AAV vectors (rAAV.hCNGA3, AAV2/8-hCARp.hCNGB3) via subretinal injection (SRI).	Phase I/II trials: NCT02610582 (rAAV.hCNGA3), NCT02935517 (ongoing). CNGB3 trials: NCT03001310, NCT02599922.	Gene replacement therapy using AAV vectors to deliver functional copies of defective genes.	Initial trials show safety and potential efficacy. Further research is needed to confirm long-term benefits.	[266,267]

**Table 6 ijms-26-04078-t006:** Ongoing Clinical Trials in Innovative Retinal Therapies.

Disease/Condition	Investigational Therapy	Mechanism/Strategy	Phase	Reference (ClinicalTri-als.gov )
Age-Related Macular Degeneration (AMD)	OpRegen^®^ (Lineage Cell Therapeutics)	Replacement treatment with allogeneic RPE cells	Phase 1/2a	NCT02286089
	NEI iPSC-RPE Patch	RPE patch made from autologous iPSCs for geographic atrophy	Early Phase 1	NCT04339764
Retinitis Pigmentosa (RP)	GS030 (GenSight Biologics)	Restoring light sensitivity in retinal ganglion cells using optogenetic treatment	Phase 1/2	NCT03326336
	hRPC Therapy (jCyte)	Injection of human retinal progenitor cells to maintain eyesight	Phase 2b	NCT03073733
Stargardt Disease	SAR422459 (Sanofi)	*ABCA4* gene delivery using gene therapy using an AAV vector	Phase 1/2	NCT01367444
X-Linked Retinitis Pigmentosa (XLRP)	AGTC-501 (AGTC)	AAV-mediated gene therapy targeting the *RPGR* gene	Phase 1/2	NCT03316560
Choroideremia	BIIB111 (Biogen, formerly Night Star)	Targeting the *CHM* gene with AAV-based gene therapy	Phase 3	NCT03496012

## 6. Conclusions

Advances in stem cell therapy, gene editing, and molecular medicine offer promising prospects for retinal regeneration and improved visual quality of life. To mitigate concerns like immunological rejection and teratogenicity, techniques involving iPSCs and MSCs require further refinement. Enhancing the efficiency and selectivity of exosome and extracellular vesicle-based therapies can enable targeted drug delivery to injured retinal tissues. Additionally, exploring small molecules and their delivery routes is crucial for optimizing retinal cell differentiation and developing more potent regenerative treatments. The successful clinical application of these cutting-edge therapies could revolutionize the treatment of retinal degenerative diseases. By addressing the underlying causes of retinal cell loss and damage, these treatments may halt disease progression and restore vision. Preclinical and clinical studies are essential to establish their safety, efficacy, and long-term benefits. Ultimately, the integration of stem cell and gene therapies into standard ophthalmic care could provide new hope for patients with currently incurable optic nerve and retinal conditions, improving their quality of life and reducing the societal burden of visual impairment.

### Simple Summary

Visual impairment is a significant global health issue affecting 2.2 billion individuals, with at least 1 billion of these cases being preventable. Current treatments manage symptoms or slow progression without addressing root causes. Devastating and irreversible vision loss is associated with retinal degenerative diseases, drawing attention to regenerative medicine-based therapeutic approaches for vision protection and restoration. Stem cell therapies show promise in replacing damaged retinal cells and reducing inflammation. Extracellular vesicles and exosomes derived from stem cells also protect retinal cells and promote repair by delivering therapeutic molecules. Bioactive molecules and growth factors can protect retinal cells from degeneration, thus prolonging vision. Gene therapy for inherited retinal diseases can correct genetic defects or introduce healthy genes and needs to be further explored for efficacy and safety. While challenges such as delivery methods and immune rejection remain, these innovative approaches could revolutionize treatment, potentially halting or reversing vision loss. Combining these therapies may provide a more comprehensive approach, improving the quality of life for millions affected by retinal degenerative diseases. Continued basic research followed by clinical trials is essential to ensure the safety and effectiveness of stem cell and gene therapy before widespread adoption.

## Figures and Tables

**Figure 1 ijms-26-04078-f001:**
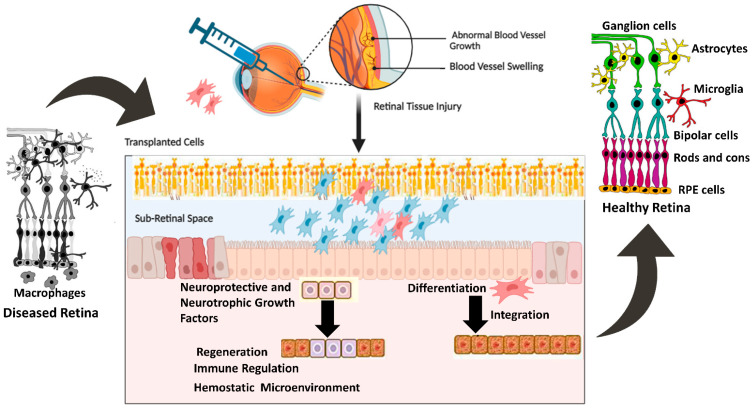
Mechanisms of cellular integration and neurotrophic factors section and their effect on retinal regeneration. The figure was designed with BioRender.

**Figure 2 ijms-26-04078-f002:**
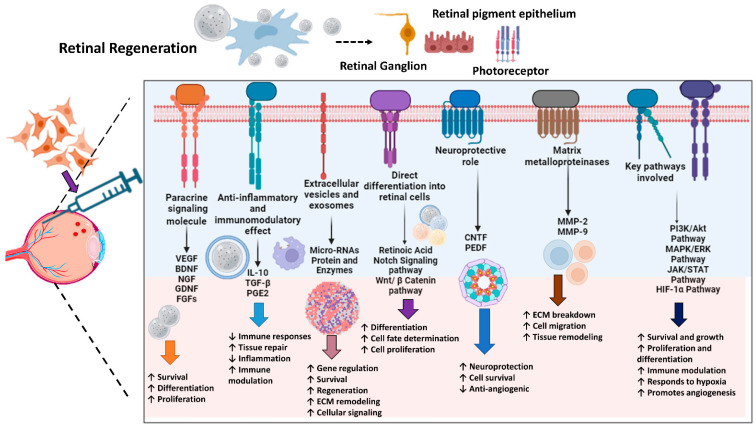
Mechanisms of Stem Cell-Mediated Retinal Regeneration. The schematic diagram represents different signaling mechanisms and their effects on retinal regeneration. The arrows (**↓** or **↑)** up or down indicate up or down regulations, respectively. The figure was designed with BioRender.

**Figure 3 ijms-26-04078-f003:**
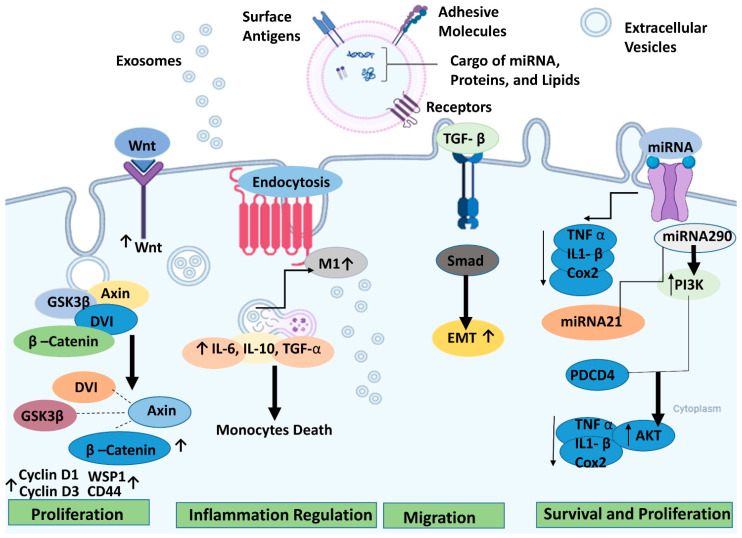
Potential underlying mechanisms of extracellular vesicles/exosomes produced from stem cells for retinal degeneration and the different signaling pathways involved. The arrows (**↓** or **↑)** up or down indicate up or down regulations, respectively. The figure was designed with BioRender.

**Figure 4 ijms-26-04078-f004:**
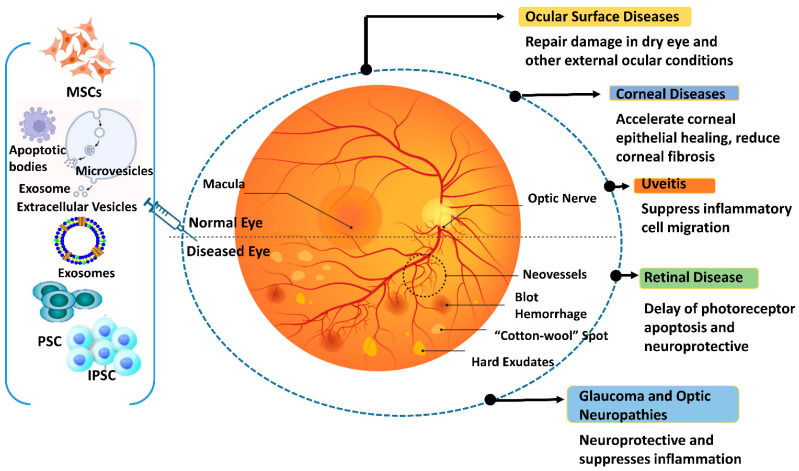
Different functions of stem cell-derived exosomes are seen in ocular disorders. The figure was designed with BioRender.

**Figure 5 ijms-26-04078-f005:**
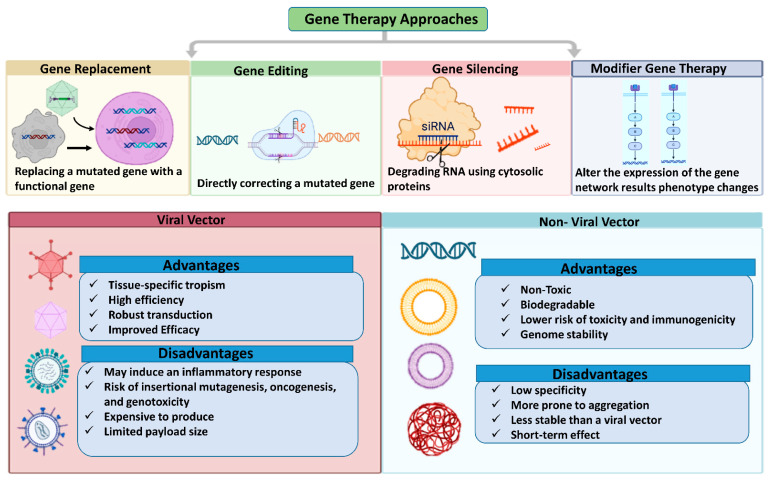
Gene therapy approaches for genetic abnormalities correcting a disease's gene or knocking down an overexpressed gene. The figure was designed with BioRender.

**Table 1 ijms-26-04078-t001:** Cell Sources and their Applications in Retinal Regeneration.

Category	Source	Key Characteristics	Applications/Findings	Challenges	References
Ocular Tissues					
NR-Derived Progenitor Cells	Neural Retina (NR)	Mainly RPCs and Müller cells	Enhanced integration of transplanted RPCs with the host retina. Retinal-specific markers are expressed by mouse and human RPCs.	Inadequate assimilation into the outermost nuclear layer. Mammals with restricted endogenous regeneration.	[99,100]
Müller Cells	NR	Cover the whole neuronal retina; astrocytes and oligodendrocytes perform CNS functions.	Retina regenerates in fish and birds. Retinal neurons may be produced in vitro by human Müller cells.	Restricted growth in mature animals. Require certain signaling pathways to function.	[101,102]
RSCs from CMZ or CB	Ciliary Marginal Zone (CMZ) or Ciliary Body (CB)	Clonally propagated differentiated into retinal cell types.	Able to produce photoreceptors. High-efficiency differentiation potential.	Solely the expression of early neural markers.	[103,104]
Multipotent Cells from RPE	RPE	Early development produces lens and retina. State of multipotency under specific circumstances.	Possibility of replacing retinal cells in treatment.	Reduced pliability as one ages.	[105,106,107,108]
Multipotent Cells from IPE	Iris Pigment Epithelium (IPE)	Potential for neurogenesis. Express markers Sox2, Nestin, and Pax6.	Mature into photoreceptors upon transplantation or co-cultivation.	Difficult to differentiate into every facet of the RPE function.	[109,110]
Non-Ocular Tissues					
ESCs	Embryos	Originating from the blastocyst’s inner cell mass. May develop into distinct kinds of retinal cells.	RPE produced from ESCs shields photoreceptors. Native RPE and hESC-RPE have comparable morphologies and functions.	Potential of carcinogenesis. Ethical issues. Rejection without suppressing immunity.	[111,112]
iPSCs	Adult Somatic Cells	Produced by Yamanaka Factors. Able to distinguish between photoreceptors and RPCs.	RGCs in the retina are produced by iPSCs. Clinical investigations indicate potential for RPE produced from hiPSCs.	Genetic abnormalities that may exist in autologous iPSCs. Rejection by the immune system.	[113,114]
NSCs or NPCs	Neural Tissue	Progenitor cells from CNS..	Survival and integration in the host retina. Able to develop into retinal cells.	Limitations on fate because of brain origin. Restricted population size.	[115]
MSCs	Bone Marrow (BMSCs)	Secrete anti-inflammatory cytokines. Differentiate into neural lines.	Treatment for retinal degeneration might potentially involve BMSC transplantation. Clinical studies for disorders of the retinal vessels.	It is uncertain how they will survive and integrate over time.	[97,98,116]

**Table 3 ijms-26-04078-t003:** Chronological List of Studies on the Role of EVs in RP, Retinopathy, and PVR.

Type of Study	Model	Findings	References
Retinitis pigmentosa			
In vivo	RP mouse model	Transport of the related protein is altered by the dominant *P347S* mutation in the rhodopsin gene.	[194]
In vivo	RP mouse model	Mutations in *TULP1* impact EV homeostasis.	[195]
In vitro/ex vivo	Human RPE cells	RPE cells emit αB-crystallin through exosomes, which PRs absorb under oxidative stress to enhance their defense.	[196]
In vivo	RP mouse model	When the *PDE6* mutation is present, the action of EVs leads to the degeneration of PRs.	[197]
In vivo	RP mouse model	Retinal degeneration is influenced by vesicles that are positive for CD9 and CD81.	[198]
In vivo	Frog (X. laevis)	Micro vesicles from deteriorating PRs are internalized by RPE cells.	[199]
In vitro/in vivo	HEK293T, EXOSC2-mutated B cells, keratinocytes, D. melanogaster	Mutations in *EXOSC2* impact the autophagy process.	[200]
Ex vivo	Human retina	In diseased retina, exosomes and α-synuclein are co-localized.	[201]
In vitro	HEK293T, mIMCD3 cells	Mutations in *PDE6D* impact the trafficking of proteins in EVs.	[202]
Retinopathy			
In vitro/in vivo	BV-2 cells, mouse model of oxygen-induced retinopathy	Exosomes produced by microglia shield PRs from harm.	[203]
In vivo	MNU-mice model	RPE cell exosomes can bring PRs’ equilibrium back.	[165]
In vitro/in vivo	HUVEC, HRMECs, C57BL/J and CD-1 mice, SD rats	In pathological settings, the transfer of anti-angiogenic agents via exosomes prevents angiogenesis.	[204]
Proliferative vitreoretinopathy			
In vitro	ARPE-19	ExomiRs cause EMC to occur.	[205]
In vitro/ex vivo/in vivo	RPE from human eyes, animal model of PVR	TSPAN4-positive RPE vesicles aid in the formation of PVRs.	[151]
Ex vivo	Exosomes from VH of PVR patients and controls	Proteins associated with inflammation, epithelial/mesenchymal transition, cellular proliferation, and connective tissue expansion are abundant in exosomes linked to PVR.	[206]

RP: Retinitis Pigmentosa, EVs: Extracellular Vesicles, RPE: Retinal Pigment Epithelium, PRs: Photoreceptors, TULP1: TUB-like Protein 1, PDE6: Phosphodiesterase 6, CD: Cluster of Differentiation, HEK293T: Human Embryonic Kidney 293T Cells, EXOSC2: Exosome Component 2, BV-2: Microglia Cell Line, MNU: N-Methyl-N-Nitrosourea, HRMECs: Human Retinal Microvascular Endothelial Cells, HUVEC: Human Umbilical Vein Endothelial Cells, ARPE-19: Adult Retinal Pigment Epithelial Cell Line, EMC: Epithelial/Mesenchymal Transition, PVR: Proliferative Vitreoretinopathy, VH: Vitreous Humor, TSPAN4: Tetraspanin 4.

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
