# Peer review of "Rekindling Vision: Innovative Strategies for Treating Retinal Degeneration"

_ijms, 2025, doi:10.3390/ijms26094078_

Round 1

Reviewer 1 Report

Comments and Suggestions for Authors

Comments on the Quality of English Language

The quality of the language used in the total manuscript does not reach the standards for publication and needs a major revision.

Author Response

Reviewer 1.

This study integrates cutting-edge fields including stem cell therapy, gene editing, and molecular medicine, demonstrating their potential for treating retinal degenerative diseases. It proposes potential research directions and technical refinements, offering valuable guidance for future studies. The work cites extensive recent research findings across all aspects of retinal degeneration treatment. It suggests that the authors have a strong understanding and command of this field. While, there are some points need to be fixed.

We appreciate the in-depth review from the reviewer, the point highlighted were addressed in the revised manuscript.

  1. The quality of the language used in the total manuscript does not reach the standards for publication and needs a major revision.

We have improved the language standard in the revised manuscript. Additionally, we requested the language and editing services of the MDPI. We hope that the reviewer will find the revised version improved.

  1. The paper lacks depth in describing certain critical technical aspects while discussing multiple therapies. This may hinder readers' comprehensive understanding of methodologies and results. The authors should provide more comprehensive descriptions and diagrams for critical experimental procedures. This will enhance understanding and make it easier for readers to replicate the experiments.

We have improved the manuscript new table 6 for ongoing clinical trials has been added. Additionally, a section on gene editing has been added. A lay summary has been incorporated to improve the understanding. We hope that reviewers will now find the manuscript improved.

  1. The note "Both authors contributed equally" is ambiguous and lacks specificity regarding their equal contribution.

The sentence has been removed. Hope reviewer will find it now with clarity.

  1. Abstract conflates inherited/acquired diseases (e.g., glaucoma vs. retinitis pigmentosa). Retinal degenerative diseases include inherited disorders (e.g., retinitis pigmentosa) and acquired disorders (e.g., AMD, diabetic retinopathy).

The abstract is now revised. We hope that the reviewer will find it improved.

  1. Figure 1 and figure 2 lack in-text callouts.

All the figures quality is improved. We hope that the reviewer will find the revised figures with more clarity.

  1. Inconsistent table formatting such as column alignment.

The table is improved. These tables will be formatted in the final version and will be presented with clarity.

  1. Outdated references and missing key advances such as CRISPR-Cas9, the authors should include 2023-2025 literatures such as "Prime editing for IRDs (Li et al., 2024, Nat Biotechnol)

We have added a section on CRISPR-Cas 9. Additionally, we have updated the references and added the reference as suggested by the reviewer.

  1. Certain techniques mentioned, such as embryonic stem cell use, may raise ethical controversies. Additional discussion is needed regarding ethical review processes and public acceptance.

This is a review article and we compiled the literature. The ethical aspect of these studies is beyond the scope of the manuscript.

  1. Inconsistent terms ("extracellular vehicles" vs. "extracellular vesicles").

These terms are corrected throughout the manuscript.

  1. Missing "Institutional Review Board Statement" and "Data Availability Statement."

We have added this information in the revised manuscript.

  1. Define abbreviations at first use (e.g., "mesenchymal stem cells (MSCs)").

The corrections have been made throughout the manuscript.

  1. Italicize gene symbols (e.g., "CEP290").

The corrections have been made throughout the manuscript.

Reviewer 2 Report

Comments and Suggestions for Authors

The manuscript reads well as far as scientific steadiness is concerned; however, the content is all over the place. It is very hard to streamline the topic of actual investigation, leveraging the literature compiled. Moreover, the topic of 'retinal degradation treatment' is not novel and there are several reviews available on the subject. The authors need to select an appropriate topic for the review and revise the content accordingly, only then will it be possible to actually perform an in-depth scientific review, based on how the prune the current manuscript.

--------------------------------------------------------------------
In the current review, the authors have discussed the various strategies for treating retinal degeneration. Though the review is well-written, the topic lacks any novelty as there are several reviews are already available on this topic. It would greatly benefit the authors if they could streamline the review and highlight the novelty in this compilation. Some other considerations to make are suggested as follows:

  1. Most of the figures are well made. However, the font size is too small in several figures. Also, some parts of the figures seem to be separate from the rest of the figure, either due to poor quality or improper merging. Please use appropriate image preparation software for generating the figures and improving their quality.
  2. Please mention whether the studies described were performed in vitro, in vivo, or in human subjects. This information is very important to understand the gravity of the work and literature that is compiled. The review can be restructured to highlight the models used for the studies.
  3. Please use a separate section on the ongoing clinical trials for the innovative retinal therapies ongoing in human patients, highlighting the disease associated, to provide more depth to the study.

Author Response

Reviewer 2.

The manuscript reads well as far as scientific steadiness is concerned; however, the content is all over the place. It is very hard to streamline the topic of actual investigation, leveraging the literature compiled. Moreover, the topic of 'retinal degradation treatment' is not novel and there are several reviews available on the subject. The authors need to select an appropriate topic for the review and revise the content accordingly, only then will it be possible to actually perform an in-depth scientific review, based on how the prune the current manuscript.

--------------------------------------------------------------------
In the current review, the authors have discussed the various strategies for treating retinal degeneration. Though the review is well-written, the topic lacks any novelty as there are several reviews are already available on this topic. It would greatly benefit the authors if they could streamline the review and highlight the novelty in this compilation. Some other considerations to make are suggested as follows:

We respect the opinion of the Reviewer 2, but we (the authors) and Reviewer 1 don’t agree with the opinion of Reviewer 2. The following comments are from Reviewer 1 about the quality of the manuscript.

“This study integrates cutting-edge fields including stem cell therapy, gene editing, and molecular medicine, demonstrating their potential for treating retinal degenerative diseases. It proposes potential research directions and technical refinements, offering valuable guidance for future studies. The work cites extensive recent research findings across all aspects of retinal degeneration treatment. It suggests that the authors have a strong understanding and command of this field.”

  1. Most of the figures are well made. However, the font size is too small in several figures. Also, some parts of the figures seem to be separate from the rest of the figure, either due to poor quality or improper merging. Please use appropriate image preparation software for generating the figures and improving their quality.

All the figures quality is improved. We hope that the reviewer will find the revised figures with more clarity.

  1. Please mention whether the studies described were performed in vitro, in vivo, or in human subjects. This information is very important to understand the gravity of the work and literature that is compiled. The review can be restructured to highlight the models used for the studies.

We have compiled these studies in table form, and this information is added there.

  1. Please use a separate section on the ongoing clinical trials for the innovative retinal therapies ongoing in human patients, highlighting the disease associated, to provide more depth to the study.

A separate table (Table 6) is added for the ongoing clinical trials in the revised manuscript.

Round 2

Reviewer 1 Report

Comments and Suggestions for Authors

I am satistifed with the revision. No other questions.

Reviewer 2 Report

Comments and Suggestions for Authors

The manuscript is acceptable in current form